
# Impacts of black carbon on the formation of advection-radiation fog during a haze pollution episode in eastern China

Qiuji Ding[1,2,3], Jianning Sun[1,2,3], Xin Huang[1,2,3], Aijun Ding[1,2,3], Jun Zou[1,2,3], Xiuqun Yang[1,2,3], and Congbin Fu[1,2,3]

[1] Joint International Research Laboratory of Atmospheric and Earth System Sciences, Nanjing University, Nanjing, China
[2] School of Atmospheric Sciences, Nanjing University, Nanjing, China
[3] Collaborative Innovation Center of Climate Change, Jiangsu Province, China

*Correspondence to*: Jianning Sun (jnsun@nju.edu.cn) and Xin Huang (xinhuang@nju.edu.cn)

**Abstract.** Aerosols can not only participate in fog formation by acting as condensation nuclei of droplets but also modify the meteorological conditions such as air temperature and moisture, planetary boundary layer height (PBLH) and regional circulation during haze event. The impact of aerosols on fog formation, yet to be revealed, can be critical in understanding and predicting of fog-haze event. In this study, we used the Weather Research and Forecasting model coupled with Chemistry (WRF-Chem) to investigate a heavy fog event during a multiday intense haze pollution episode in early December 2013 in the Yangtze River Delta (YRD) region in eastern China. Using the WRF-Chem model, we conducted four parallel numerical experiments to evaluate the roles of aerosol-radiation interaction (ARI), aerosol-cloud interaction (ACI), black carbon (BC) and none BC (non-BC) aerosols in the formation and maintenance of the heavy fog event. Only when the aerosols' feedback processes are considered can the model well capture the haze pollution and the fog event. We find that the ARI dominates this fog-haze episode while the effects of ACI are negligible. Our analyses shows that BC plays a more important role in fog formation than non-BC aerosols. The dome effect of BC leads to an increase of air moisture over the sea by reducing PBLH and weakening vertical mixing, thereby confining more water vapor in the near-surface layer. The strengthened daytime onshore flow by a cyclonic wind anomaly, induced by contrast temperature perturbation over land and sea, transports moister air to the YRD region, where the suppressed PBLH and weakened daytime vertical mixing maintain the high moisture level. Then the heave fog forms due to the surface cooling at night in this region. This study highlights the importance of anthropogenic emissions in the formation of advection-radiation fog in the polluted coastal areas.

## 1 Introduction

Fog, defined as visible water droplets or ice crystals suspended in the near-surface atmosphere (Gultepe et al., 2007), has been considered as hazardous weather that affects airline, marine transport, and high-way traffic (S. Niu et al., 2010). The economic loss caused by low visibility due to fog becomes comparable to that of tornadoes or even hurricanes (Gultepe et al., 2007). Fog often forms in polluted atmosphere when the air moisture is relatively high, so the relationship between aerosols



and fog droplets is complex in the polluted area (Guo et al., 2015). On one hand, aerosols can act as cloud condensation nuclei (CCN) during the activation and diffusion growth of fog droplets (Kuroiwa, 1951), and determine the property of droplets such as its microstructure (Hudson, 1980) or chemical composition (Munger et al.,1983; Wang et al., 2011; Yang et al., 2012). On the other hand, the fog water can provide a medium for aqueous-phase reactions and transform primary

pollutants into secondary aerosols (Graedel et al., 1985; Dall'Osto et al., 2009; Xie et al., 2015; Wang et al., 2016; Cheng et al., 2016). As an important removal process of aerosols, wet scavenging effect of fog can also significantly reduce aerosol concentration in the atmosphere (Yuskiewicz et al., 1998; S. Niu et al., 2010). Thus, the interrelations between fog and aerosols are crucial for better understanding of the formation and evolution of both fog and haze pollution.

Due to the intensive emission and unfavorable meteorological conditions, China has been undergoing a number of severe

haze pollution events during last two decade (D. Wu et al., 2005; Ding et al., 2013a; Zhao et al., 2013; Y. Zhang et al., 2016; Huang et al., 2018). Many studies investigated the responses of mesoscale circulation to the high aerosol loading during air pollution events over eastern China (Ding et al., 2013b; J. Wang et al., 2014; H. Wang et al., 2015; Gao et a., 2015; Ding et al., 2016a; Huang et al., 2018). Recently, black carbon (BC) has attracted a lot of attention (Bond et al., 2013), since it can significantly influence the planetary boundary layer (PBL) structure (Ding et al., 2016a; Wang et al., 2018; Huang et al.,

2018). By absorbing the incident shortwave radiation and warming the atmosphere, BC can stabilize the PBL, which leads to heavy air pollution in megacities (Ding et al., 2016a; Huang et al., 2018; Wang et al., 2018), and even changes the weather such as precipitation in heavily polluted area (Ding et al., 2013b; Huang et al., 2016).

Meteorological conditions are important to the persistent severe fog and haze event over eastern China (Zhang et al., 2014; Y. Zhang et al., 2016). The winter fog over eastern China is typically advection-radiation fog (Lin et al., 2017). It results from

20 radiative cooling of moist air which is advected from any nearby large water body (Ryznar, 1977). Since that aerosols can impact meteorological field, such as changing the distribution of temperature and moisture and modifying the atmospheric stratification, their influences on the fog formation should not be neglected. Bott et al. (1991) pointed out that the radiative effect of absorbing aerosols can modify the structure of nocturnal PBL and favor the formation of fog in an urban environment. F. Niu et al. (2010) studied the impact of high-loading aerosols on the increase of winter fog and found that the

25 warming effect of absorbing aerosols can promote the fog formation over southern China by weakening the eastern Asian monsoon circulation and increasing the aerosol concentration. Since the dynamics and/or thermodynamics of the interaction between atmosphere and aerosols are complex, the details about the impacts of aerosols on fog process have not been well understood yet. The eastern China has a relatively high BC emission rate and a high loading of BC (Qin and Xie, 2012; Andersson et al., 2015), which has been found to significantly influence the PBL structure and to enhance haze pollution

(Ding et al., 2016a). However, the impact of BC on fog, an important PBL phenomenon in winter, has been rarely investigated by existing studies.

During 1-10 December 2013, eastern China was suffering from a multi-day severe haze and fog event with high PM$_{2.5}$ concentrations and low visibility (Ding et al., 2016a; Sun et al., 2017). On 7 December, heavy fog event was recorded by many meteorological stations in the Yangtze River Delta (YRD) region. In this work, we study the impacts of aerosols,



especially the BC, on this fog event and the associated meteorological conditions based on numerical simulations using the Weather Research and Forecasting model coupled with Chemistry (WRF-Chem). In section 2, the WRF-Chem model, observational data and experiment design are introduced. In section 3, the simulation results are evaluated by observational data, the different roles of Aerosol Radiation Interaction (ARI), Aerosol Cloud Interaction (ACI), BC and non-BC aerosols

in affecting the meteorological conditions and fog processes are analyzed, and the mechanism of enhanced moisture advection induced by radiative effect of BC is investigated. Section 4 provides summaries and conclusions.

## 2 Data and Methodology

### 2.1 WRF-Chem model and configuration

In numerical simulation of fog, not only the accurate meteorological background conditions are needed (Gultepe, et al.,

2007), but also the high grid resolution and adequate representation of underlying surface are vital to resolve the physical process and patchy structure of fog (Steeneveld et al., 2015). Therefore the mesoscale weather model has been extensively applied to simulate the fog episodes (Shi et al., 2010; Lin et al., 2017). However, in the previous studies the effects of ARI or ACI on fog are rarely considered. Hence, in this study we use the mesoscale non-hydrostatic WRF-Chem model (version 3.8.1) to investigate the heavy fog-haze event over eastern China during 1-10 December 2013. WRF-Chem is an on-line

coupled air quality model which can simultaneously calculate the meteorology field, the transport, mixing, and chemical transformation of trace gas and aerosols, and the effects of ARI and ACI (Grell et al., 2005; Fast et al., 2006; Chapman et al., 2009).

The physical and chemical parameterization schemes adopted in this study are listed in Table 1. The Morrison double-moment microphysics scheme is linked with cloud chemistry, washout of trace gases, and explicit aerosol aqueous-phase

chemistry (Morrison et al., 2009). The Grell-3D cumulus scheme which allows subsidence in neighbouring columns is coupled with cloud chemistry and tracer transport (Grell and Devenyi, 2002). The RRTMG radiation scheme (Iacono et al., 2008), which is linked with aerosol optical properties, is applied to both longwave and shortwave radiations. The Noah Land Surface scheme is used together with land-use data from the Moderate Resolution Imaging Spectro-radiometer (MODIS) (Chen and Dudhia, 2001), where the urban surface process was simulated by the single-layer urban canopy model (UCM)

(Chen et al., 2011). The PBL scheme of Mellor-Yamada-Nakanishi-Niino level-2.5 (MYNN2) (Nakanishi and Niino, 2006) is a 1.5-order and local turbulence closure scheme, which can estimate the PBL height (PBLH) according to the prognostic turbulence kinetic energy (TKE). The gas-phase mechanism of Model for Ozone And Related chemistry Tracers (MOZART) contains a detailed description of tropospheric chemistry (Emmons et al., 2010). The aerosol scheme of Model for Simulating Aerosol Interactions and Chemistry (MOSAIC) (Zaveri et al., 2008) treats major aerosol species including sulfate,

sulfonate, nitrate, chloride, carbonate, ammonium, sodium, calcium, black carbon, primary organic mass, and other inorganic matters etc. Secondary organic aerosol formation based on Lane et al. (2008) is included in MOSAIC with a volatility basis set which provides an empirical representation of the aging and volatility of the organic aerosol and its precursors. Four



sectional size bins are employed to represent particle size distribution in the MOSAIC scheme, where aerosols are assumed to be internally mixed within each bin. The photolysis rates are simulated with Fast Tropospheric Ultraviolet-Visible (FTUV) model (Tie et al., 2003).

Anthropogenic emission is Hemispheric Transport of Air Pollution (HTAP v2.2) monthly sector-specific emission inventory
for the year 2010 at a spatial resolution of 0.1°×0.1° grid maps. Figure 1 shows the horizontal distribution of BC emission over eastern China. This dataset, which combines nationally reported emissions and regional scientific inventories, is obtained from Emission Database for Global Atmospheric Research (EDGAR) (Janssens-Maenhout et al., 2015). Biogenic emission is calculated online by Model of Emissions of Gases and Aerosol from Nature (MEGAN) (Guenther et al., 2006) while dust emission is simulated online using the Georgia Tech/Goddard Global Ozone Chemistry Aerosol Radiation and
Transport (GOCART) parameterization (Zhao et al., 2010).

The period of simulation is from 30 November to 10 December in 2013 with the first 24 hours as the spin-up time. The simulation domain, as shown in Fig. 1, covers eastern China and the surrounding area by a horizontal resolution of 15 km. There are 26 vertical levels extending from the ground surface up to 50 hPa, with about 15 levels below the altitude of 1.5 km. The lowest level is located at approximately 60 m above the ground. The initial and boundary conditions of
15 meteorological field are provided by National Centers for Environmental Prediction (NCEP) with a horizontal resolution of 1°x1° for every 6 hours. The chemical initial and boundary conditions are provided by MOZART-4 global model output at 1.9°×2.5° for WRF-Chem (Emmons et al., 2010).

## 2.2 Experiment design

In WRF-Chem, ARI is simulated through coupling the aerosol optical properties with the radiation transfer calculation. The
20 radiative properties of aerosols are estimated based on Mie theory. ACI is simulated by coupling the cloud physics and MOSAIC aerosol scheme. Based on Kohler theory, CCN is calculated as a function of aerosol number concentrations and updraft velocity according to the parameterization of aerosol's activation by Abdul-Razzak and Ghan (2002). Since that both ARI and ACI play important roles in affecting meteorology, parallel numerical experiments were designed to quantify their individual effect. The control experiment (EXP_CTL), in which both ARI and ACI are turned on, represents the scenario
including the complex physical and chemical processes between aerosols and the atmosphere. Two experiments, one turns off ARI (EXP_NORAD) and the other turns off both ARI and ACI (EXP_NOAER), are conducted to simulate the scenarios in absence of aerosol radiative effect and without any aerosol effects, respectively. Thus, the difference between EXP_CTL and EXP_NORAD represents the effect of ARI (EF_ARI). The difference between EXP_NORAD and EXP_NOAER is expected to represent the effect of ACI (EF_ACI). The roles of BC and non-BC aerosols in modifying meteorological
conditions can be very different and need to be treated separately. Hence, another experiment, in which BC is removed from the anthropogenic emission (EXP_NOBC), is conducted to identify the relative contribution of different aerosol components. The difference between EXP_CTL and EXP_NOBC is regarded as the effect of BC (EF_BC), and the difference between EXP_NOBC and EXP_NOAER considered as the effect of non-BC (EF_NBC). The total effect of aerosols (EF_TOT) is



derived from the difference between EXP_CTL and EXP_NOAER. To make it more clear, the brief description on the aforementioned experiments and the effects of different aerosol feedback processes are listed in Table 2.

## 2.3 Observational data

The observational data on both meteorology and air quality are employed to evaluate the model performance. Hourly measurement data of air temperature and relative humidity (RH) at the altitude of 2 meter above the ground (T2 and RH2), and wind speed at the altitude of 10 meter above the ground (WS10) at 153 meteorological stations in eastern China are collected from the National Climate Data Center (NCDC) of China. Hourly $PM_{2.5}$ concentrations are obtained from China National Environmental Monitoring Center (CNEMC). This dataset includes hourly concentrations of $SO_2$, $NO_2$, CO, $O_3$, $PM_{2.5}$ and $PM_{10}$ at 74 cities covered emission-intensive regions such as the YRD and Beijing-Tianjin-Hebei (BTH). The observational data of all sites within each city is averaged to represent the local pollution levels. The geographical positions of NCDC and CNEMC observations are illustrated in Fig. 1. The data of vertical profiles of air temperature, RH and water vapor mixing ratio (Q) in Nanjing are obtained from the atmospheric sounding dataset of Earth Observing Laboratory, National Center for Atmospheric Research (NCAR) (http://weather.uwyo.edu/upperair/sounding.html). The data of downward shortwave radiation (DSR) and the surface sensible heat flux (SHF) in north-eastern suburban of Nanjing are used to validate the simulation results. These datasets are obtained from the Station for Observing Regional Processes of the Earth System (SORPES) in the Xianlin Campus of Nanjing University (Ding et al., 2013a; Ding et al., 2016b; Zou et al., 2017).

## 3 Results and discussion

### 3.1 Validation of simulated meteorological variables, $PM_{2.5}$ concentration and Fog events

The interaction between atmosphere and aerosols influences both meteorological conditions and the distribution of aerosols (Ding et al., 2013b; Ding et al., 2016a; Huang et al., 2016). In this section, evaluations of meteorological field and $PM_{2.5}$ concentration are carried out for the EXP_CTL, the experiment including full physical and chemical processes. The statistics of T2, RH2, WS10 and $PM_{2.5}$ mass concentration between the simulation and observation are summarized in Table 3. Mean observation (Mean Obs.), simulation (Mean Sim.), bias (MB), root mean square error (RMSE) and correlation coefficient (R) are calculated for T2, RH2, WS10 over 153 NCDC stations and $PM_{2.5}$ concentrations over 74 CNEMC sites. Overall, the model reproduced a satisfying representation of meteorological conditions with acceptable biases. T2 is simulated with a cold bias of –0.76°C, a RMSE of 2.93 and a correlation of 0.93. RH2 simulation has a negligible MB of –0.18, a RMSE of 16.35 and a correlation of 0.69. WS10 is overestimated with a MB of +0.61 m s$^{-1}$. As for surface pollutants, the model successfully reproduces $PM_{2.5}$ concentration with a relatively small MB of +1.54 μg m$^{-3}$. The mean simulated and observed $PM_{2.5}$ concentrations are 150.72 μg m$^{-3}$ and 149.18 μg m$^{-3}$ respectively. The simulated $PM_{2.5}$ concentrations agree well with the observations at 74 sites over eastern China during this haze event (Fig. 2). However, $PM_{2.5}$ temporal variations are not well reproduced with a relatively large RMSE of 124.33 μg m$^{-3}$ and relatively small correlation of 0.40. This is probably




because that the measured PM$_{2.5}$ concentrations are averaged in a city scale, which doesn't match with the exact grid of WRF-Chem.

Previous observational study has demonstrated that the reduced DSR and SHF are subject to the increase of aerosol concentration, which consequently leads to suppression of daytime PBL development (Zou et al., 2017). We also compared the simulated DSR and SHF in different experiments with the observations at the SORPES station in Nanjing. The comparisons show that DSR and SHF in EXP_CTL agreed well with the observational data, however, they were overestimated by EXP_NORAD, EXP_NOAER and EXP_NOBC (Fig. S1). These results suggest that the radiative effect of aerosols can be reproduced reasonably when both the ARI and ACI processes are included in the simulations.

The evolution of fog-haze event during 1-10 December 2013 over eastern China is illustrated in four stages as shown in Fig. 3 and Fig. 4. First, during the daytime of 2 December, the northern part of eastern China was under the control of southwest wind, and the horizontal distribution of air pollutant shows that PM$_{2.5}$ concentration was relatively high in BTH while relatively low in YRD (Fig. 3a). The air quality measurements show that surface PM$_{2.5}$ concentration was high in Zhengzhou and Nanjing (about 150 μg m$^{-3}$), and relatively low in Beijing (less than 100 μg m$^{-3}$) (Fig. 4). Second, on 5 December large amount of PM$_{2.5}$ was transported to the mid-eastern China from BTH by the north wind. Meanwhile the west wind transported PM$_{2.5}$ to the Yellow Sea (Fig. 3b). In Nanjing, the surface concentration of PM$_{2.5}$ exceeded 300 μg m$^{-3}$ during the daytime with the maximum value of 370 μg m$^{-3}$ at 12:00 LST (Fig. 4). Third, on 6 December the heavy haze pollution in YRD region continued and the PM$_{2.5}$ concentration in Nanjing stayed around 300 μg m$^{-3}$. As shown in Fig. 3c, the PM$_{2.5}$ concentration in the southern part of this region reached a very high level with the maximum value of about 400 μg m$^{-3}$. Moreover, the eastward wind transported a large amount of PM$_{2.5}$ from YRD region into the northern part of East China Sea. Finally, the fog-haze event in eastern China dissipated on 9 December (Fig. 3d) due to the arrival of strong cold air from northern China. Figure 4 shows that the model can successfully capture the main feature of fog-haze evolution. However, PM$_{2.5}$ and BC concentrations were substantially underestimated during the occurrence of fog in Nanjing. This discrepancy may be caused by an overestimation of fog-induced wet deposition of PM$_{2.5}$ and BC or the PBL height during the mixed fog/haze events. A relatively larger bias between observation and simulation in PM$_{2.5}$ than BC during the daytime of 7 December also indicates a possible enhancement of the formation of secondary aerosols through aqueous phase or heterogeneous reactions (Xie et al., 2015; Zheng et al., 2015; Wang et al., 2016; Cheng et al. 2016).

On 7 December, fog events were recorded by many meteorological stations in the YRD region. Simulation results show that this fog episode was successfully predicted by EXP_CTL (fog is simulated by the numerical model when RH reaches 100% and liquid water accumulated near surface). Figures S2a and S2b show the comparisons between the observed and simulated RH2 and T2 at Lukou site which locates at 36 km south of Nanjing. They agree very well before the onset of fog and during the fog event. The fog began at 00:00 LST on 7 December when RH2 reached 100% (Figs. S2a). Before this time, T2 decreased in the early night on 6 December due to radiative cooling, which led to an increase of RH2 and finally made the water vapor saturated. During the fog event, T2 remained almost unchanged, because the radiative cooling can be compensated by the latent heat release in the process of fog formation during the night, while after sunrise the heating effect





of solar radiation can be exhausted by the evaporation of fog droplets. The fog dissipated at about 12:00 LST on 7 December. The EXP_CTL can capture the evolution of T2 and RH2 during this fog episode (Figs. S2a and S2b) and also the vertical profiles of Q and RH in the PBL at 08:00 LST on 7 December (Figs. S2c and S2d).

## 3.2 Impact of aerosols on the formation and evolution of the fog event

5 ### 3.2.1 Effects of ARI and ACI

High loading of BC and non-BC aerosols can substantially reduce the solar radiation reaching the Earth's surface, resulting in significant change in surface meteorological conditions (Ding et al., 2013b; Gao et al., 2015; H. Wang et al., 2015; Ding et al., 2016a; Huang et al., 2018). Meanwhile, aerosol particles can act as CCN, and ACI can modify the solar radiation which also leads to the change of surface meteorological conditions (K. Wang et al., 2015; Liu et al., 2016). To disentangle their 10 individual contribution, Figure 5 shows the average distributions of EF_TOT, EF_ARI and EF_ACI on the DSR, the near surface temperature (Tns) and the near surface water vapor mixing ratio (Qns) in the daytime (08:00 - 17:00 LST) during 1-10 December. It shows that the magnitude of EF_ACI on DSR, Tns and Qns was negligible small, suggesting an insignificant role of ACI during this fog-haze episode. Since the winter haze in eastern China often associates with high pressure system, subsidence of air mass can lead to less cloud cover condition which consequently weakens the effect of ACI 15 (B. Zhang et al., 2015). Figure 5b and 5e show that $\Delta$DSR and $\Delta$Tns had different patterns in spatial distribution. The distribution of $\Delta$DSR followed the pattern of PM$_{2.5}$ concentration shown in Fig. 2. In the YRD region, $\Delta$DSR was more than –75 W m$^{-2}$, while in the adjacent offshore region $\Delta$DSR ranged from –25 to –50 W m$^{-2}$ (Fig. 5b). However, as shown in Fig. 5e, the EF_ARI resulted in a cooling effect over land but a warming effect over sea. The maximum cooling effect occurred in the eastern part of the YRD region, where the magnitude of $\Delta$Tns reached –1.0 °C. The maximum warming effect took 20 place in the East China Sea with the magnitude of $\Delta$Tns as high as +0.8 °C. The different patterns of $\Delta$DSR and $\Delta$Tns were attributed to the competition of warming effect of BC in the atmospheric and cooling effect of the extinction caused by both BC and non-BC aerosols. The distribution of $\Delta$Tns suggests that the cooling effect exceeded the warming effect over the land while the inverse situation occurred over the sea. To our knowledge, by scattering and absorbing the solar radiation, aerosols reduce DSR reaching the Earth's surface, which leads to the reduction of surface temperature. However, the 25 atmospheric heating due to absorbing aerosols (e.g., BC) would somehow offset by the surface dimming through turbulent heat exchange. Comparatively, the sea surface has larger capacity to storage heat than the land, and hence the near surface temperature over the sea was less sensitive to the reduction of incoming solar radiation and would be more effectively increased by the aerosol warming effect (K. Wang et al., 2015). This is why the cooling effect was found over the land while warming effect over the sea. Figure 5g shows a significant EF_ARI on Qns in the YRD region and the East China Sea. In 30 this area, the increase of the near surface water vapor mixing ratio was associated with the occurrence of an advection-radiation fog on 7 December.



For the fog episode on 7 December, Fig. 6 shows the simulation results of liquid water content (LWC) near the surface and the temporal change of vertical distribution of LWC in EXP_CTL and EXP_NOAER. When considering the feedback processes of all aerosols, the spatial distribution of near surface LWC in EXP_CTL shows that at 08:00 LST the dense fog was located mainly in the YRD region (Fig. 6a). The temporal change of vertical distribution of LWC in Nanjing shows that the fog episode began at 00:00 LST and dissipated at 12:00 LST (Fig. 6b). In the EXP_NOAER which excluded the effects of both ARI and ACI, the fog only appeared in a very small area located at the upper edge of YRD region (Fig. 6c), and no fog occurred in Nanjing (Fig. 6d). These discrepancies confirm that the fog formation was promoted by ARI and ACI. However, the results in Fig. 5 suggest that the EF_ACI is negligible small in this fog-haze episode. It can be expected that the widely spread fog in YRD region was mainly induced by the EF_ARI. The mechanism for the fog formation will be discussed later.

### 3.2.2 Contributions from BC and non-BC aerosols

The physical processes of fog development are subject to various factors such as radiative cooling, moisture advection, evaporation, and even turbulent mixing. Our simulations show that during the haze-fog event ARI played a dominant role, which was caused by the radiative effects of BC and non-BC aerosols. To investigate the roles of different types of aerosols, as well as their total effect, we evaluated the contribution of each feedback process on DSR, SHF, surface water vapor flux (QFX), Tns, Qns, RHns and PBLH based on different experiments. How to calculate the contribution of the corresponding feedback process is explained in Table 2. For example, the contribution of BC was derived from the difference between EX_CTL and EX_NOBC. Calculations were only conducted in the significantly influenced area, i.e., the YRD region, at 14:00 LST during 1-10 December. The quantities were averaged over this area (the grid points of waterbody are excluded). The relative contributions are illustrated in Fig. 7.

The highest concentrations of BC (13.56 μg m$^{-3}$) and non-BC aerosols (174.42 μg m$^{-3}$) appeared on 6 December (Fig. 7a), and the corresponding contributions of BC and non-BC aerosols to the decrease of DSR are –86.39 W m$^{-2}$ and –78.98 W m$^{-2}$ respectively (Fig. 7b). Although the concentration of BC was much smaller than that of non-BC aerosols, the cooling effect of BC on the incoming solar radiation at the surface was comparable to that of non-BC aerosols. As a result, the decreases of SHF induced by BC and non-BC aerosols were comparable, and the maximum decreases occur on 6 December (Fig. 7c). The relative contributions and variation patterns were similar in terms of QFX (Fig. 7d). However, the response of near surface atmospheric temperature to the radiative effect of BC was quite different to that of non-BC aerosols. As shown in Fig. 7e, the non-BC aerosols had a negative contribution to Tns (i.e., to decrease the near surface temperature), while BC had a positive impact on Tns (i.e., to increase the near surface temperature). This is because BC would trap parts of the solar energy in the atmosphere that was supposed to reach the ground surface. Our simulations suggest that, during this heavy air pollution event in the YRD region, the warming effect of BC can exceed its cooling effect and the net effect is to increase the air temperature. A decrease of temperature directly led to an increase of relative humidity. Thus, Fig. 7h shows that non-BC aerosols increased RHns. However, this figure also shows that BC decreased RHns before 6 December but have a positive



contribution to RHns during 6-7 December. The positive contribution of BC to RHns on 6-7 December implies that RH was influenced by not only the local temperature perturbation but also other forcings. Figure 7f shows that the BC has a positive contribution to Qns during 6-7 December, suggesting that the effect of BC was to increase the moisture in this area by other processes. Figure 7g shows that the non-BC aerosols always have a negative contribution to PBLH, because the shadowing

effect reduces the SHF and consequently decreases the PBLH (Zou et al. 2017). Meanwhile, BC also had a negative impact on PBLH during 6-7 December, although BC aerosols tended to increase the near surface temperature in this period. The reason is that during the heavy air pollution event the heating effect of BC was more effective in the upper air than in that of the lower part of PBL, which stabilized the atmospheric stratification and suppressed the PBL development in the daytime. This effect has been well documented and named as the dome effect of BC (Ding et al., 2016; Wang et al., 2018).

To better understand the roles of BC and non-BC aerosols on the fog episode, we calculated their individual contributions to Tns, Qns, RHns and LWCns from 14:00 LST on 6 December to 14:00 LST on 7 December in YRD region. Figure 8a shows that the total effect of BC and non-BC aerosols was to decrease the near surface air temperature, although BC tended to increase the temperature on 6 December. It means that Tns in EXP_CTL is lower than that in EXP_NOAER and the cooling effect of aerosols was favourable for fog formation. However, Fig. 8b shows that BC significantly increased Qns in the

afternoon and early night on 6 December while the contribution of non-BC aerosols was quite small. Significant increase of water vapor induced by BC led to a relatively high RHns (above 60%) in the afternoon. Due to the decrease in temperature, the RHns increased, and then the fog formed around 00:00 LST on 7 December (Fig. 8c). After the onset of fog, the Qns decreases (Fig. 8b) and the LWCns increased (Fig. 8d), because the condensation process changed water vapor into liquid water during fog development. As shown in Figure 8d, BC played a dominant role in fog formation. Spatially, the

contribution of BC is also significantly larger than that of non-BC aerosols on enlarging the fog area (Fig. S3). Overall, the above results suggest that in this fog episode the most important factor is the increase of water vapor in this area and BC plays an important role in the fog formation. However, as pointed out previously, the local effect of BC on meteorological conditions alone cannot explain the increase of water vapor. It can be expected that the significant increase of water vapor in YRD region is caused by moisture advection.

**3.3 BC-induced moisture advection between ocean and land**

Besides the radiative cooling of the surface and the adjacent atmosphere, a long-duration and thick-layered fog often requires other conditions such as turbulent mixing and advection (Wobrock et al., 1992). When not considering BC aerosol (EXP_NOBC), in the coastal area of the north of YRD region the wind was from the Yellow Sea towards the land, and the onshore flow brought the moist air to the north of YRD region, producing an isolated high moisture area at 14:00 LST on 6

December (See the central part around 32˚N in Fig. 9b). At the same time in the EXP_CTL which included BC's effect, the moisture over the Yellow Sea was increased and the onshore flow was enhanced. This led to an enlarged and amplified high moisture center in the northern part of the YRD region (the central part in Fig. 9a). The comparison between EXP_CTL and EXP_NOBC suggests that BC enhanced the moisture advection from the Yellow Sea to the northern part of the YRD region.




Figure 9 also shows that the southward wind in this area was enhanced by BC, which increased the transport of the moister air into YRD region where the fog formed that night.

Fig. 10a shows that high value of BC concentration covered the eastern part of the YRD region and the nearby offshore area at 14:00 LST on 6 December (denoted by the isolines), which led to a warming effect on near-surface layer atmosphere in this area. Then the warmed atmosphere resulted in a low-pressure perturbation in this area, which consequently led to a cyclonic anomaly in wind field (Fig. 10b). This cyclonic anomaly strengthened the onshore wind speed in the north-eastern part of YRD region because the background wind here was approximately in the same direction. Meanwhile, the surface layer over the nearby sea was moisturized due to the heating effect of BC (Fig. 10d). Thus, the moisture advection from the sea to the land was mainly promoted by BC. To better understand the enhancement of moisture advection induced by BC, we calculated the difference of horizontal moisture flux in the surface layer between EXP_CTL and EXP_NOBC. The horizontal moisture flux $\boldsymbol{MF}$ is defined as the production of specific humidity q (with a unit of g m$^{-3}$) and velocity $\boldsymbol{V}$. We calculate the magnitude of $\boldsymbol{\Delta MF}$ as $|\boldsymbol{\Delta MF}|=|\boldsymbol{MF_{BC}}| - |\boldsymbol{MF_{NOBC}}|$ (i.e., the difference in magnitude of horizontal moisture flux between the two cases), and denote the direction of $\boldsymbol{\Delta MF}$ by an arrow, whose direction is the direction of $\boldsymbol{\Delta V}$ and whose length represents the magnitude of $\boldsymbol{\Delta V}$. Figure 10c shows that the area with a large value of $|\boldsymbol{\Delta MF}|$ was located at the northern edge of YRD region and the nearby offshore area at 14:00 on 6 December, and the enhanced advection was onshore (only the arrows with the magnitude larger than 3 m s$^{-1}$ are denoted in this figure). Since the wind direction in this area was mainly southward, the enhanced moisture advection would move into YRD region in the later time. To make it more clear, a cartoon, which demonstrates BC's perturbation on moisture advection and the fog evolution, is provided in the Supplemental Information. The cartoon is linked by a sign in the figure caption of Figure S4, which shows the state at the beginning time 10:00 on 6 December.

Another role of BC is to depress the PBLH in the daytime. Figure S5 shows a vertical cross-section at the location denoted in Fig. 10d. The largest increase of air temperature caused by the heating effect of BC occurred in the upper part of PBL over the land, and the PBLH over Nanjing was only half of that when excluding BC's effect. This is consistent with the previous study by Ding et al. (2016a), in which the dome effect of BC, which stabilizes the PBL and suppresses PBL development in the daytime, has been discovered for the first time. A decreased PBLH restricted the water vapor within a shallow layer near the ground surface, which can increase the moisture in the PBL (if the PBL is well developed, the water vapor will be spread vertically in a deeper layer, which can reduce the water vapor content at the ground surface). The stabilized PBL weakened the vertical mixing, which made near-surface air moister (Fig. S5), which promoted the accumulation of water vapor near surface in the daytime when the moister air was transported to the land. Therefore, the fog can form in the whole YRD region in the night when the role of BC was included. Meanwhile, a stabilized PBL and a decreased PBLH caused by the dome effect of BC can also explain the increase of moisture in the surface layer over the sea. We provide more evidences in Fig. S6.




## 4 Conclusions

During 1-10 December 2013 eastern China underwent a haze pollution episode, and on 7 December a heavy fog, associated with a haze event with high concentration PM$_{2.5}$, occurred over the YRD region. In order to better understand the impacts of aerosols on fog formation and the associated meteorological conditions, we applied numerical modeling with the online-coupled air quality model WRF-Chem to simulate this fog-haze event and to explore the main mechanisms of haze-fog interaction. By conducting four parallel numerical experiments, we analyzed the effects of ARI, ACI, BC and non-BC aerosols. We find that BC played the key role in this fog-haze event by producing favourable local and regional meteorological conditions for the formation of advection-radiation fog in the YRD region.

First of all, evaluation of model performance shows that WRF-Chem can well capture the air pollution process and the fog episode. The simulated meteorological conditions, such as temperature, relative humidity and wind speed in the surface layer, agree well with the observations. However, when excluding the interaction between meteorology and aerosols, there are large biases in meteorological conditions and the fog cannot be reproduced. Our results support that intense atmospheric pollution can modify weather in previous studies (e.g., Ding et al. 2013b; Huang et al., 2016).

Comparison between ARI and ACI shows that the effect of ARI was dominant over the effect of ACI. The reason may be that during this haze pollution episode there was little cloud and ACI was very weak. Further analyses are made to reveal the different roles of BC and non-BC aerosols in the fog formation. The results show that they both contribute to the fog formation. However, non-BC aerosols have a relatively small contribution by decreasing temperature in the surface layer, while BC has a larger contribution by significantly increasing moisture in this layer. The radiative effect of BC was to reduce the local surface evaporation, which can reduce the air moisture, but BC-induced circulation anomalies could increase moisture by enhanced advection between land and sea.

The mechanism of BC-induced moisture advection was explored based on detailed analysis of modeling results from different experiments. High loading of BC in the YRD region and the nearby offshore area exerted a dome effect on the PBL, which directly increased moisture over the sea by reducing PBLH and weakening vertical mixing, because more water vapor was confined in the lower part of a shallow PBL. The regional warming effect of BC on the air temperature led to a low-pressure perturbation in this area, and consequently resulted in a cyclonic anomaly in wind field. The cyclonic anomaly enhanced the onshore flow, which brought more moisture from the sea into the YRD region in the north during the daytime. Then the increased moisture helped to form a heavy advection-radiation fog during night time. Meanwhile, the dome effect of BC produced lower PBLH and weaker vertical mixing in the daytime. These were favourable for maintaining high moisture in the daytime and the formation of fog in the night.

This study highlights the complexity of interaction among ARI, regional circulation and PBL, in which the effect of BC is very important, and suggests that in heavy polluted area the numerical weather prediction model should include the feedback processes, especially those induced by radiatively active species like BC.



**Author contributions.** JS designed the study. JS, AD, XY and CF developed the measurement station to record meteorology and atmospheric chemical composition. QD and XH conducted model simulations. QD and ZJ analyzed the data. JS and AD prepared the paper with contributions from all authors.

**Competing interests.** The authors declare that they have no conflict of interest.

**Acknowledgements** The work is supported by National Natural Science Foundation of China (91544231) and Ministry of Science and Technology of China (2016YFC0200500). We are grateful to the High-Performance Computing & Massive
Data Center (HPC&MDC) of School of Atmospheric Science, Nanjing University for supporting the numerical calculations in this paper.

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



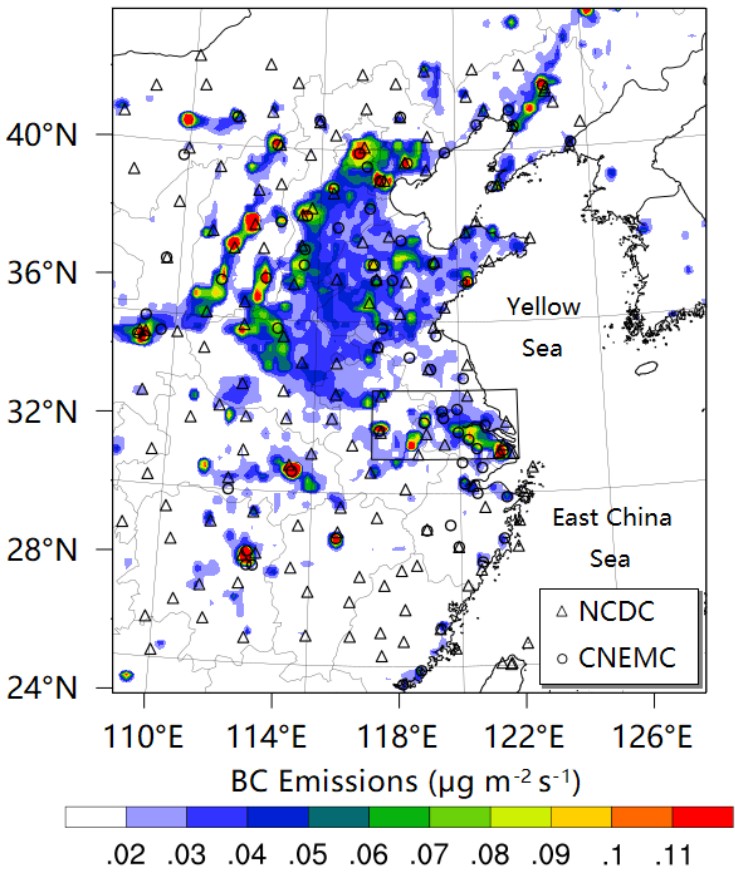

**Figure 1: A map showing the WRF-Chem model domain, BC emission and observation sites for meteorological parameters and air pollutants. Color filled contours represent the BC emission from EDGAR-HTAP dataset. The triangles represent the locations of 153 meteorological stations over eastern China from NCDC, and the circles represent the locations of the 74 cities from where the pollutant observations are recorded. The rectangle denotes the YRD region.**



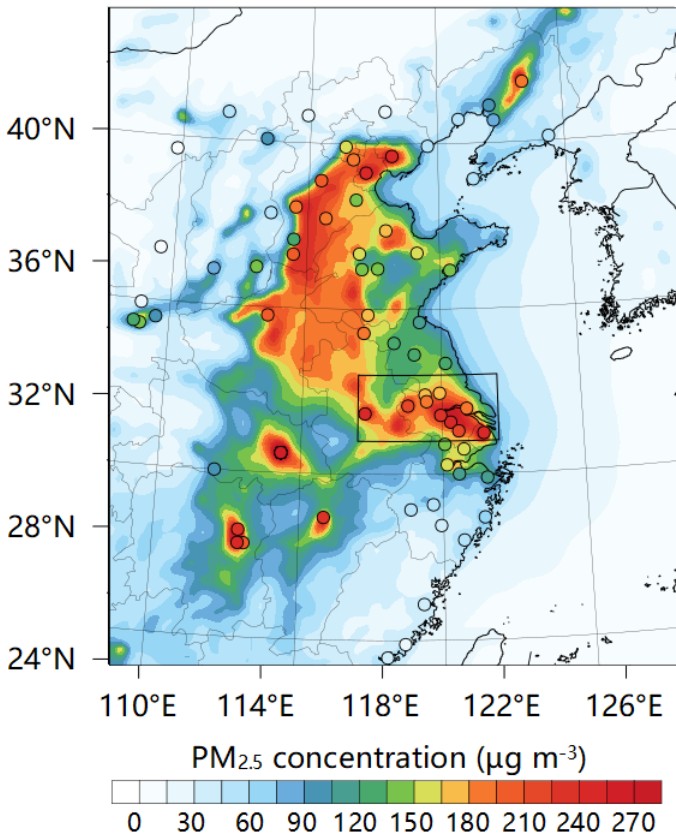

**Figure 2.** Spatial distribution of simulated (colored area) and observed (color coded dots) mean PM$_{2.5}$ concentrations during 1-10 December 2013 over eastern China. The black rectangle denotes the YRD region.




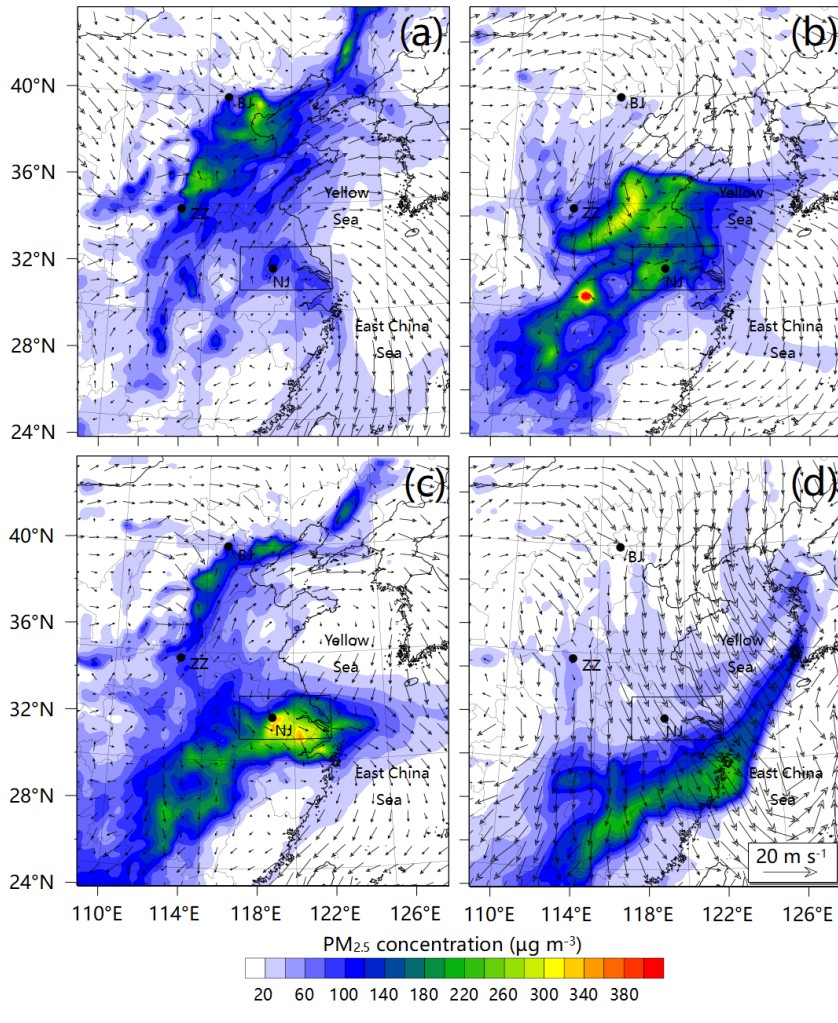

**Figure 3. Spatial distribution of PM$_{2.5}$ mass concentration (µg m$^{-3}$) and wind field (m s$^{-1}$) near the ground surface from EXP_CTL at 14:00 on 2 (a), 5 (b), 6 (c) and 9 (d) of December. The black rectangle denotes the YRD region, and the black dots represent locations of Beijing (BJ), Zhengzhou (ZZ) and Nanjing (NJ).**




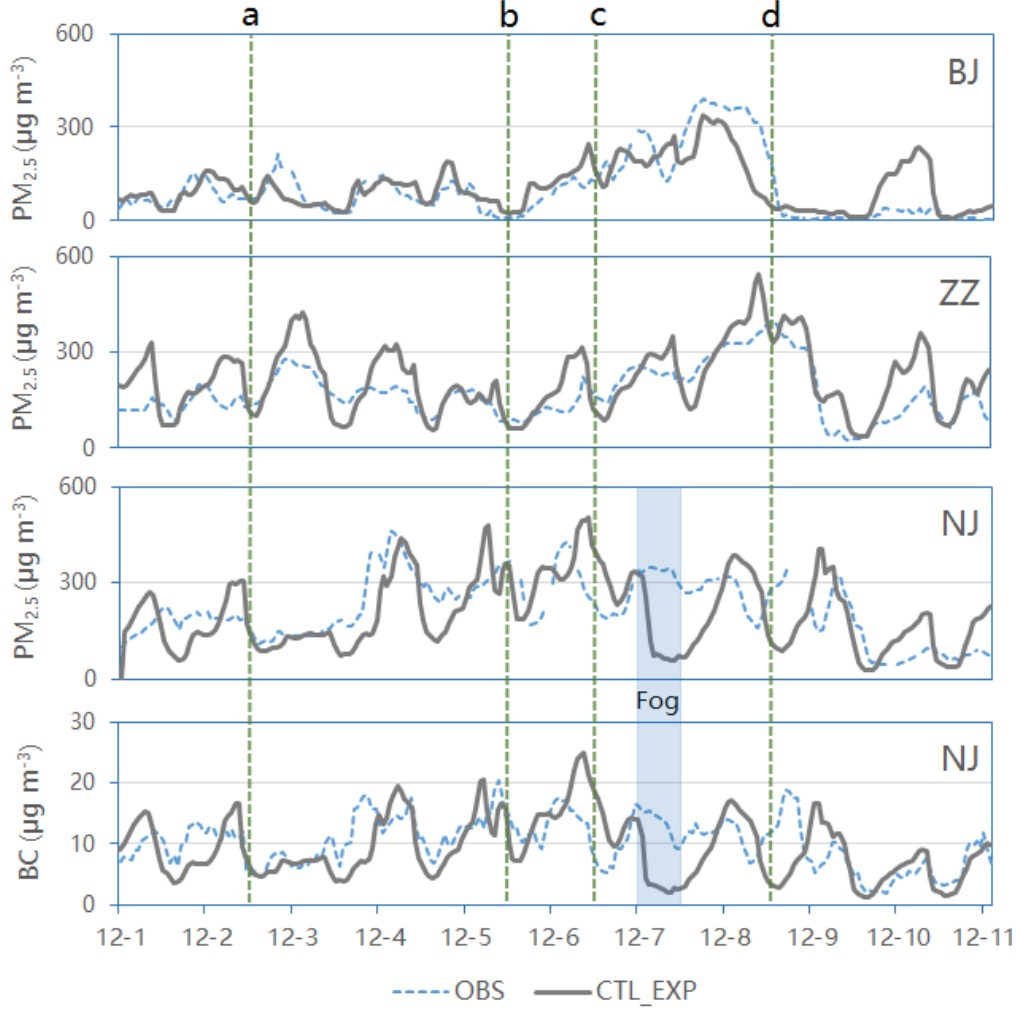

**Figure 4. Temporal variation of surface aerosol concentration in Beijing (BJ), Zhengzhou (ZZ), and Nanjing (NJ). Observational data (blue dashed line) of PM$_{2.5}$ are from CNEMC and observational data of BC are from the SORPES station in NJ. The simulated results are from EXP_CTL. The vertical green dashed lines with label *a – d* denote the corresponding time as in Figs. 3a-d. The blue area denotes the duration of fog episode in YRD region.**

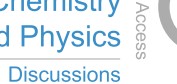

**Figure 5.** Average changes of DSR (upper), $T_{ns}$ (middle) and $Q_{ns}$ (lower) induced by EF_TOT (left), EF_ARI (middle) and EF_ACI (right) in the daytime (08:00 – 17:00 LST) during 1-10 December 2013. The subscript "ns" represents "near the surface", i.e., the surface layer. The physical properties are calculated from the simulated results at the lowest grid level (approximately 60 m above the ground).



**Figure 6. The distribution of LWC$_{ns}$ from EXP_CTL (a) and EXP_NOAER (c) at 08:00 LST on 7 December, and the vertical distribution of LWC at Nanjing from EXP_CTL (b) and EXP_NOAER (d) varying from 14:00 LST on 6 December to 14:00 LST on 7 December. The dashed line denotes the PBLH.**





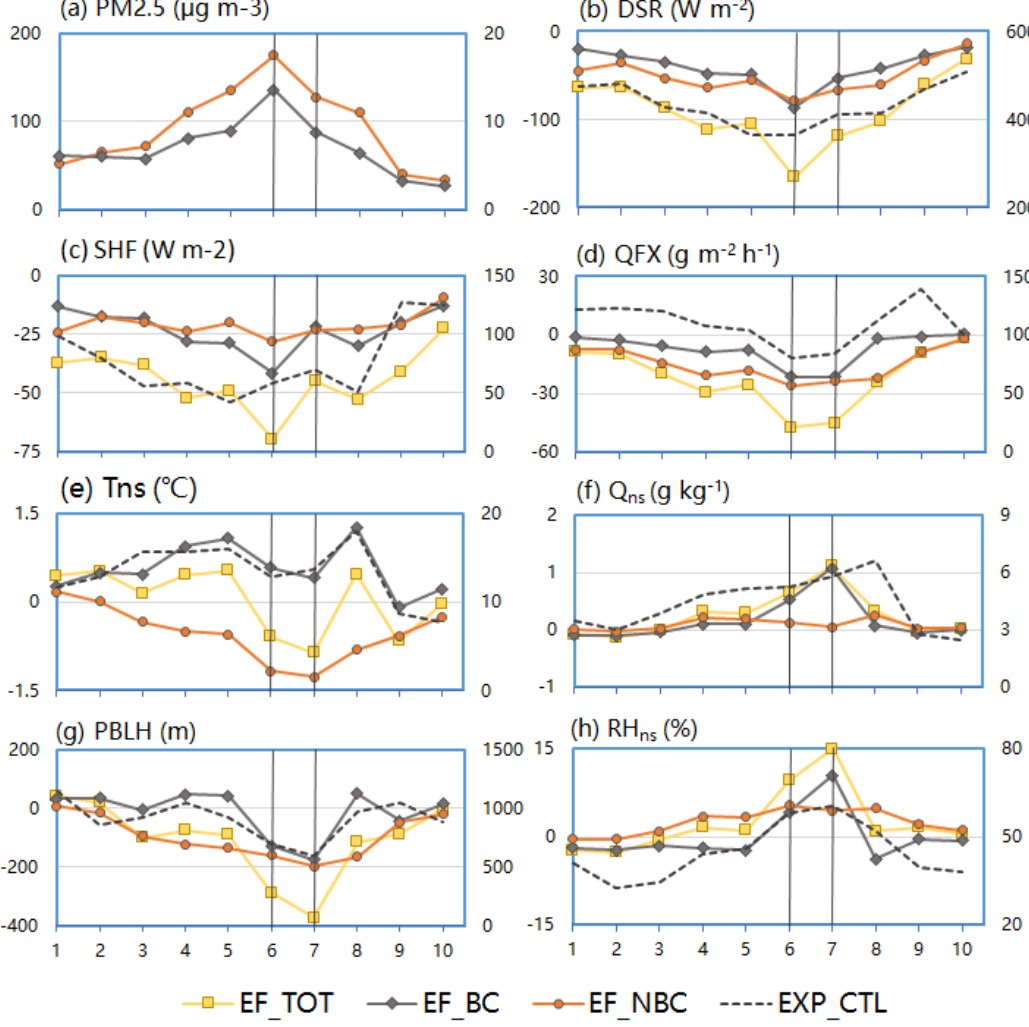

**Figure 7. The contributions from EF_TOT, EF_BC, EF_NBC to the change in PM$_{2.5}$ concentration, DSR, SHF, QFX T$_{ns}$, Q$_{ns}$, PBLH and RH$_{ns}$ at 14:00 LST during 1-10 December 2013 (the magnitude is denoted by the left axis). The corresponding values from EXP_CTL (the magnitude is denoted by the right axis) are plotted for reference. The data are calculated from the simulated results at the lowest grid level over the YRD region, as marked in Fig. 1.**



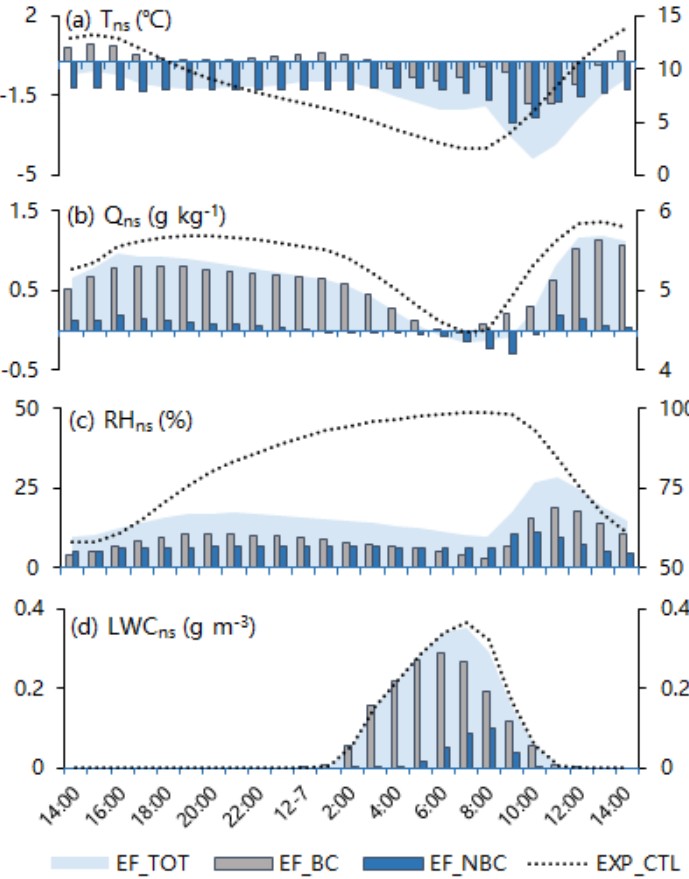

**Figure 8. The temporal evolution of the contributions of EF_TOT, EF_BC, EF_NBC to the change of $T_{ns}$ (a), $Q_{ns}$ (b), $RH_{ns}$ (c) and $LWC_{ns}$ (d) from 14:00 LST on 6 December to 14:00 LST on 7 in the YRD region (the magnitude is denoted by the left axis), and the results from EXP_CTL (the magnitude is denoted by the right axis).**




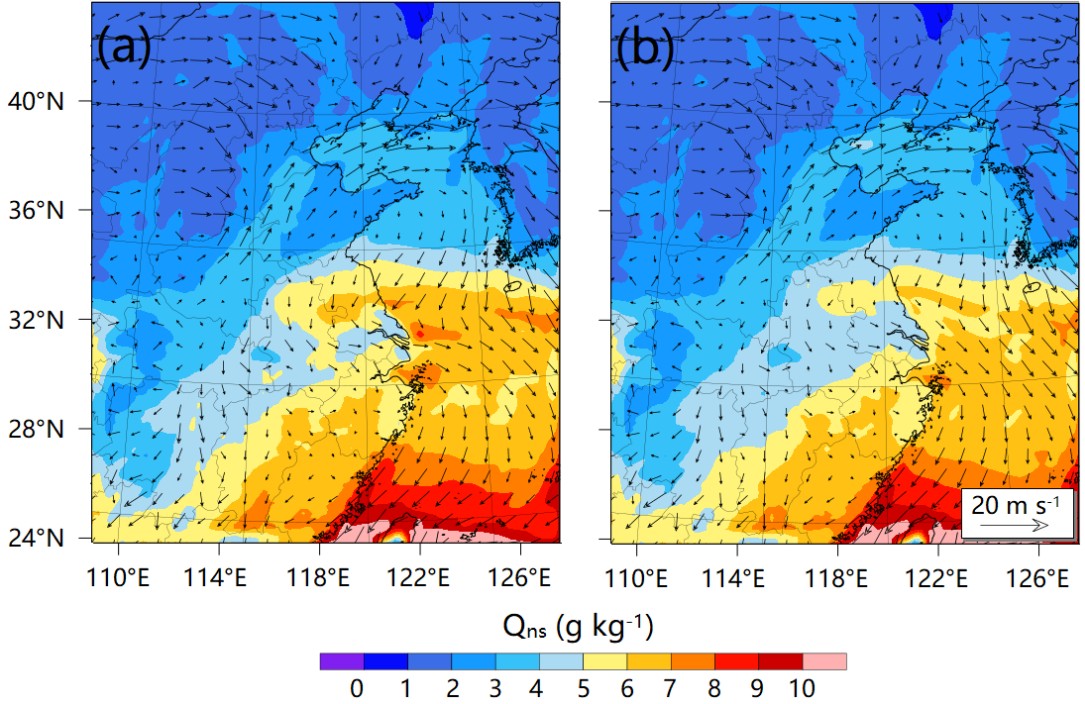

**Figure 9.** Spatial distribution of $Q_{ns}$ (g kg$^{-3}$) and $V_{ns}$ (m s$^{-1}$) from EXP_CTL (a) and EXP_NOBC (b) at 14:00 LST on 6 December.





**Figure 10. BC-introduced increase of air temperature (a), low-pressure perturbation and cyclonic anomaly in wind field (b), increase of moisture advection (c), and increase of water vapour mixture ratio (d) in the surface layer at 14:00 LST on 6 December. Arrows shown in (c) are the BC-induced velocity change with the magnitude larger than 3 m s⁻¹, which are used to represent the direction of enhanced horizontal moisture flux. The line in (d) denotes the location where the cross-section of BC-induced increase of temperature and moisture (as well as the PBLH in different experiments) is plotted in Fig. S5, and the rectangle denotes the offshore area, in which the average profiles of potential temperature and water vapor mixture ratio in EXP_CTL and EXP_NOBC are calculated and plotted in Fig. S6.**



**Table 1.** WRF-Chem modelling configuration options

| Physics | Selected Schemes |
|---|---|
| Microphysics | Morrison double-moment |
| Cumulus | Grell 3D ensemble |
| Radiation | RRTMG (LW&SW) |
| Land Surface | NOAH Land Surface Model |
| PBL | MYNN 2.5 |
| Urban | Urban Canopy Model |
| **Chemistry** | **Selected Schemes** |
| Gas-phase | MOZART |
| Aerosol | 4-bins MOSAIC with VBS |
| Photolysis | Madronich F-TUV |
| Dust | GOCART with AFWA |
| Biogenic | MEGAN 2 |



**Table 2.** Experiments and the effects of different aerosol feedback processes

| Experiments | Descriptions | |
|---|---|---|
| EXP_CTL | full emission and aerosol feedbacks | |
| EXP_NORAD | turn off aerosol radiative effect | |
| EXP_NOAER | turn off aerosol radiative and cloud effect | |
| EXP_NOBC | remove the emission of BC | |
| **Effect** | **Descriptions** | **Calculations** |
| EF_TOT | aerosol total impact | EXP_CTL – EXP_NOAER |
| EF_ARI | effect of ARI | EXP_CTL – EXP_NORAD |
| EF_ACI | effect of ACI | EXP_NORAD – EXP_NOAER |
| EF_BC | impact of BC | EXP_CTL – EXP_NOBC |
| EF_NBC | impact of non-BC | EXP_NOBC – EXP_NOAER |



**Table 3.** Statistics for evaluation of model (EXP_CTL) performance against observations

| variables | Samples | Mean Sim. | Mean Obs. | MB | RMSE | R |
|---|---|---|---|---|---|---|
| T2 (°C) | 14588 | 5.16 | 5.93 | -0.76 | 2.93 | 0.93 |
| RH2 (%) | 14586 | 60.66 | 60.84 | -0.18 | 16.35 | 0.69 |
| WS10 (m s$^{-1}$) | 13395 | 3.16 | 2.55 | +0.61 | 1.96 | 0.58 |
| PM$_{2.5}$ (μg m$^{-3}$) | 16201 | 150.72 | 149.18 | +1.54 | 124.33 | 0.40 |