# Peer review of "Impacts of black carbon on the formation of advection-radiation fog during a haze pollution episode in eastern China"

_Atmospheric Chemistry and Physics, 2019_

## Referee Comment (RC1) · Anonymous Referee #2 · 7 Apr 2019

This study investigates the impact of aerosol-cloud interactions (ACI) and aerosol-radiation interactions (ARI) on fog formation. The important roles of changing of advection and PBL dynamics in fog formations are revealed. It highlights the role of BC in the formation and maintenance of fog. In general, the manuscript is well organized. Thus, I suggest a minor revision before publication. The suggestions and comments are lists as following:

1. Introductions, Line 30, Page 2: I think it is not appropriated to claims "the impact of BC on fog has been rarely investigated by existing studies" after listing two references of "the impact of absorbing aerosols on fog formations". Since BC is one of the most

important absorbing aerosols.

2. Could you describe whether nudging is employed and the detailed method of nudging? Because nudging can affect the estimation of ARI.

3. As mentioned in Aerosol-Radiation-Microphysics Interactions (https://ruc.noaa.gov/wrf/wrf-chem/wrf_tutor ial_2018/AerosolInteractions.pdf, page 42), "Comparing runs with chem_opt = 8 (without cloud-borne aerosols) with chem_opt = 10 (with cloud-borne aerosols) for MOSAIC coupled to Lin microphysics does not quantify the indirect effect, since the autoconversion scheme used in the Lin microphysics scheme will be different". I'm not sure if it is the same for Morrison module? Could you describe the prescribed aerosol used in EXP_NOAER scenario?

4. Mentioned in Line 15, page 11, "Comparison between ARI and ACI shows that the effect of ARI was dominant over the effect of ACI. The reason may be that during this haze pollution episode there was little cloud and ACI was very weak." However, the role of ACI is not limited as the change of solar radiation and PBL dynamics. One of the most important ways is acting as CCN during fog formations. I'm curious if the number and radius of fog droplet are changed. Could you show some results?

5. I think some statements in abstract and conclusion are too strong as a case study, like "We find that the ARI dominates this fog-haze episode while the effects of ACI are negligible." It would be more appropriate if the statement could be limited as for specific scenarios. Further, it would be interesting to investigate under which conditions ARI is more important and under which conditions, ACI is more important in future studies. Maybe it is beyond the scope of this study.

6. I'm not sure whether the sharp decrease of PM2.5 and BC on Dec 7 in Figure 4 is due to wet removal or not. I guess the sharp decrease and rapid increase may be caused by the activation of the interstitial aerosol to the cloud-borne aerosol, and resuspension from the cloud-borne aerosol to the interstitial aerosol. And cloud-borne aerosol is not counted in CTL_EXP scenario. If so, could you check if the cloud-borne

aerosol is calculated in the optical module and discuss whether ARI is underestimated during fog episode?

---

## Referee Comment (RC2) · Anonymous Referee #1 · 7 Apr 2019

In this study, the authors examined evaluate the roles of aerosol-radiation interaction (ARI), aerosol-cloud interaction (ACI), black carbon (BC) and none BC (non-BC) aerosols in the formation and maintenance of the heavy fog event in early 15 December 2013 in the Yangtze River Delta (YRD) region in eastern China using WRF-Chem model. They found that ARI dominates this fog-haze episode while the effects of ACI are negligible. BC plays a more important role in fog formation than non-BC aerosols, inducing temperature contrast over land and sea and transports moister air to the YRD region. The topic of this study is interesting and the manuscript is well written. I would suggest publishing after addressing my comments below.

My main concern is the method to distinguish the role of BC and non-BC. First, the authors turned off ARI and ARI+ACI to separate the roles of ARI and ACI, which is reasonable. Then the authors compare the simulation with ARI+ACI turned off and with anthropogenic BC emission removed to separate to roles of non-BC and BC, which could be inappropriate. These roles were quantified using different methods and comparing them could lead to an apple and orange comparison. Without BC emission, the internal mixing state could be changed, the impact of BC on vegetation could be changed, chemistry on aerosol surface could be changed, besides the BC-radiation and BC-cloud interactions. To separate the role of BC and non-BC, I would suggest the authors do another parallel simulation with all aerosol emissions turned off. Or at least discuss the biases of the results.

Minor comments:

Page 1 Line 24: 'heave' -> 'heavy'

Page 2 Line 15: In addition to impact on PBL, recent study also found BC can change land-sea thermal contrast and weaken the East Asian winter monsoon (Lou et al., 2019), which mechanism is very similar to this study, as well as large-scale circulations (Yang et al., 2019).

Page 4 Line 26: Please clarify that how did the authors turn of ARI and ACI in the model. What variables they excluded or fixed in model?

Page 5 Line 4: Please provide the sites or data reservoir where the authors got all these data.

Figure 4: It should be days 2,5,6,"9".

Page 6 Line 22: underestimated during the occurrence of fog in Nanjing "in the model".

Page 6 Line 26: a possible enhancement of the formation of secondary aerosols through aqueous phase or heterogeneous reactions "in the real world".

Page 7 Line 23-28: I am confused that why temperature over the sea was less sensitive to the reduction of incoming solar radiation but sensitive to aerosol warming effect? Both of them are changes in radiative fluxes. Large heat storage capacity only implies that ocean is insensitive to heat or cooling effects (changes in radiative fluxes) compared to land. Instead, Lou et al. (2019) found that BC-induced heating over Bohai Sea, Yellow Sea, and East China Sea evaporated low cloud and increased high cloud, leading to larger warming over oceans east of China mainland.

Page 8 Line 30: The BC impact on surface temperature depends on vertical location of BC and also depends on models.

Page 9 Line 2: What does "other forcings " mean.

References:

Lou, S., Y. Yang, H. Wang, S. J. Smith, Y. Qian, and P. J. Rasch, Black carbon amplifies haze over the North China Plain by weakening the East Asian winter monsoon, Geophys. Res. Lett., 46, 452–460, doi:10.1029/2018GL080941, 2019.

Yang, Y., S. J. Smith, H. Wang, C. M. Mills, and P. J. Rasch, Variability and timescales in the climate response to black carbon emissions, Atmos. Chem. Phys., 19, 2405-2420, doi:10.5194/acp-19-2405-2019, 2019.

---

## Author Comment (AC1) · 12 May 2019

**Response to Referee #2**

*This study investigates the impact of aerosol-cloud interactions (ACI) and aerosol radiation interactions (ARI) on fog formation. The important roles of changing of advection and PBL dynamics in fog formations are revealed. It highlights the role of BC in the formation and maintenance of fog. In general, the manuscript is well organized. Thus, I suggest a minor revision before publication. The suggestions and comments are lists as following:*

**Response**:

We appreciated the referee to offer us these insightful suggestions. We will revise this article accordingly.

*1. Introductions, Line 30, Page 2: I think it is not appropriated to claims "the impact of BC on fog has been rarely investigated by existing studies" after listing two references of "the impact of absorbing aerosols on fog formations". Since BC is one of the most important absorbing aerosols.*

**Response**:

Accepted. We will revise "rarely investigated" to "not fully understood" in the introduction.

*2. Could you describe whether nudging is employed and the detailed method of nudging? Because nudging can affect the estimation of ARI.*

**Response**:

The application of nudging can affect the feedback from chemistry module to the meteorology. So we did not apply any nudging while simulating of ARI or ACI. We will clearly demonstrate this in the revision.

*3. As mentioned in Aerosol-Radiation-Microphysics Interactions (https://ruc.noaa.gov/wrf/wrf-chem/wrf_tutor ial_2018/AerosolInteractions.pdf, page 42), "Comparing runs with chem_opt = 8 (without cloud-borne aerosols) with chem_opt = 10 (with cloud-borne aerosols) for MOSAIC coupled to Lin microphysics does not quantify the indirect effect, since the autoconversion scheme used in the Lin microphysics scheme will be different". I'm not sure if it is the same for Morrison module? Could you describe the prescribed aerosol used in EXP_NOAER scenario?*

**Response**:

In all experiments, the cloud water mixing ratio is predicted by the Morrison schemes. In EXP_NOAER, model simulated aerosol has no impact on the cloud/fog droplet. Instead, a constant value of cloud droplet number (250 cm$^{-3}$) is used, which is a default treatment in the WRF model. In other experiments (EXP_CTL, EXP_NORAD and EXP_NOBC), the prognostic cloud droplet number is applied. The purpose of the autoconversion is to convert

cloud droplet or ice (mass and number) into snow or rain due to collision–coalescence between cloud droplets. The Lin scheme would have a different autoconversion parameterization when switching from constant value to prognostic droplet number, which is why it's not qualified for estimating the indirect effect. But in the Morrison scheme, the same autoconversion parameterization is implemented for either constant or prognostic droplet number (Yang et al., 2011). Moreover, in our result, the episode of fog-haze event is neither affected by precipitation during the early December 2013, nor the fog process are largely influenced by the autoconversion in the microphysics schemes since very little fog is predicted in the YRD region on 7 December in EXP_NOAER scenario.

*4. Mentioned in Line 15, page 11, "Comparison between ARI and ACI shows that the effect of ARI was dominant over the effect of ACI. The reason may be that during this haze pollution episode there was little cloud and ACI was very weak." However, the role of ACI is not limited as the change of solar radiation and PBL dynamics. One of the most important ways is acting as CCN during fog formations. I'm curious if the number and radius of fog droplet are changed. Could you show some results?*

**Response**:

The ARI took a dominate role in both affecting the meteorological condition and inducing the fog occurrence. If without the enhanced moisture advection induced by the ARI, the moisture level in the YRD region would be relatively unfavorable for the formation of fog, and the ACI effect of aerosols acting as the fog nuclei can be limited (under low relative humidity). As a result, the ACI has a very small impact on the liquid water content in the near surface (LWCns) of the YRD region at 10 LST on 7 December in Figure S3b (which is provided in the supplemental information of the article). Figure R1 shows the time series of the ARI, ACI and aerosols' total impact on the water vapor mixing ratio (Qns) and LWC in the near surface of the YRD region. It also suggests that during the fog episode of 7 December, the ARI took a dominate role in both introducing the Qns and LWCns. There is only a small contribution from ACI (less than 0.05 g m$^{-3}$) on enhancing the LWCns. The effect of ACI is calculated as the difference between EXP_NORAD and EXP_NOAER. In EXP_NORAD, which remove only ARI, both of the cloud water mixing ratio and number concentration is predicted. In EXP_NOAER, which remove both ARI and ACI, the cloud water mixing ratio is predicted but the number concentration is fixed to 250 cm$^{-3}$. Figure R2-R4 shows the liquid water content (LWC), number concentration and the effective radius of cloud droplet at LST 7:00 7 December 2013 in EXP_NORAD and EXP_NOAER. A small enhancement of LWCns can be seen in the northern part of the YRD region due to ACI (Figure R2). Accordingly, in EXP_NORAD, the number concentration is above 350 cm$^{-3}$ which is above the constant value of 250 cm$^{-3}$ in EXP_NOAER (Figure R3), and the effective radius of the droplet is mildly decreased (Figure R4) from EXP_NORAD to EXP_NOAER.

[Figure]

**Figure R1.** The ARI, ACI and aerosols' total impact on the near surface water vapor mixing ratio and liquid water content during 1-10 December 2013 in the YRD region.

[Figure]

**Figure R2.** Liquid water content in the near surface over the eastern China, LST 7:00 December 2013, from (a) EXP_NORAD and (b) EXP_NOAER.

[Figure]

**Figure R3.** Number concentration of cloud droplet in the near surface over the eastern China, LST 7:00 December 2013, from (a) EXP_NORAD and (b) EXP_NOAER.

[Figure]

**Figure R4.** Effect radius of droplet in the near surface over the eastern China, LST 7:00 December 2013, from (a) EXP_NORAD and (b) EXP_NOAER.

*5. I think some statements in abstract and conclusion are too strong as a case study, like "We find that the ARI dominates this fog-haze episode while the effects of ACI are negligible." It would be more appropriate if the statement could be limited as for specific scenarios. Further, it would be interesting to investigate under which conditions ARI is more important and under which conditions, ACI is more important in future studies. Maybe it is beyond the scope of this study.*

**Response:**

Accepted and thanks for the suggestions. Some modifications will be applied to avoid strong statements about the experiment result. We are also interested in the question about under which condition the ARI/ACI is more effective in affect fog formation. Although it is not easy to answer this question from one single case study, we will include this suggestion in our future work.

6. *I'm not sure whether the sharp decrease of PM2.5 and BC on Dec 7 in Figure 4 is due to wet removal or not. I guess the sharp decrease and rapid increase may be caused by the activation of the interstitial aerosol to the cloud-borne aerosol, and resuspension from the cloud-borne aerosol to the interstitial aerosol. And cloud-borne aerosol is not counted in CTL_EXP scenario. If so, could you check if the cloud-borne aerosol is calculated in the optical module and discuss whether ARI is underestimated during fog episode?*

**Response:**

Accepted. The $PM_{2.5}$ and BC that measured in our observation is dehumidified particle which include both interstitial and cloud-borne aerosols. However, the $PM_{2.5}$ and BC provide by the simulation result is the dry mass from only interstitial aerosol. The cloud-borne aerosol is processed in the CTL_EXP separately as aerosols-in-cloud-water type variables, which is not included in the Figure 4. So, we will revise the explanation for the underestimation of the $PM_{2.5}$ and BC on Dec 7 accordingly. Also, the optical effects from the cloud-borne aerosol are not directly estimated, in the optical module from the WRF-Chem that we used (version 3.8.1). In fact, many climate models tend to omit the treatment of optical effect from cloud-borne aerosol as well, because the scattering of the sunlight can be less efficient once the aerosol particle attached to the droplet (Ghan and Easter, 2006). However, Bond et al. (2013) pointed out the absorption of the BC can be increased when it is coated with non-absorbing material include water, and the neglect of this effect can influence the general estimation of the radiative forcing in global scale. However, in this study, the cloud cover is low during the fog-haze event. So, the neglect of the optical effect from the cloud-borne aerosol did not have much influence on the radiation flux or other meteorological fields. And it would not affect the fog formation since it mostly took place during nighttime. Only after the sunrise, when the fog top is heated by the solar radiation, the cloud-borne BC can absorb more sunlight, warm the ambient atmosphere and speed up the evaporation of the droplets at the top fog layer. The absence of the optical effect from cloud-borne aerosol may, to some extent, slow down the dissipation of fog. From another perspective, under the heavy fog condition, the reduction of the incoming solar radiation mostly resulted from the light extinction of fog droplets, which can overwhelm the direct ARI from cloud-borne aerosol particle.

**References :**

Bond, T. C., Doherty, S. J., Fahey, D. W., Forster, P. M., Berntsen, T., DeAngelo, B. J., Flanner, M. G., Ghan, S., Karcher, B., Koch, D., Kinne, S., Kondo, Y., Quinn, P. K., Sarofim, M. C.,

Schultz, M. G., Schulz, M., Venkataraman, C., Zhang, H., Zhang, S., Bellouin, N., Guttikunda, S. K., Hopke, P. K., Jacobson, M. Z., Kaiser, J. W., Klimont, Z., Lohmann, U., Schwarz, J. P., Shindell, D., Storlvmo, T., Warren, S. G., and Zender, C. S.: Bounding the role of black carbon in the climate system: A scientific assessment, J. Geophys. Res.-Atmos., 118, 5380–5552, https://doi.org/10.1002/jgrd.50171, 2013.

Ghan, S. J. and Easter, R. C.: Impact of cloud-borne aerosol representation on aerosol direct and indirect effects, Atmos. Chem. Phys., 6, 4163-4174, https://doi.org/10.5194/acp-6-4163-2006, 2006.

Yang, Q., W. I. Gustafson Jr., Fast, J. D., Wang, H., Easter, R. C., Morrison, H., Lee, Y.-N., Chapman, E. G., Spak, S. N., and Mena-Carrasco, M. A.: Assessing regional scale predictions of aerosols, marine stratocumulus, and their interactions during VOCALS-REx using WRF-Chem, Atmos. Chem. Phys., 11, 11951-11975, https://doi.org/10.5194/acp-11-11951-2011, 2011.

---

## Author Comment (AC2) · 12 May 2019

**Response to Referee #1**

*In this study, the authors examined evaluate the roles of aerosol-radiation interaction (ARI), aerosol-cloud interaction (ACI), black carbon (BC) and none BC (non-BC) aerosols in the formation and maintenance of the heavy fog event in early 15 December 2013 in the Yangtze River Delta (YRD) region in eastern China using WRF-Chem model. They found that ARI dominates this fog-haze episode while the effects of ACI are negligible. BC plays a more important role in fog formation than non-BC aerosols, inducing temperature contrast over land and sea and transports moister air to the YRD region. The topic of this study is interesting and the manuscript is well written. I would suggest publishing after addressing my comments below.*

**Response:**

   We are appreciated the comments, which are helpful to further improve the article.

*My main concern is the method to distinguish the role of BC and non-BC. First, the authors turned off ARI and ARI+ACI to separate the roles of ARI and ACI, which is reasonable. Then the authors compare the simulation with ARI+ACI turned off and with anthropogenic BC emission removed to separate to roles of non-BC and BC, which could be inappropriate. These roles were quantified using different methods and comparing them could lead to an apple and orange comparison. Without BC emission, the internal mixing state could be changed, the impact of BC on vegetation could be changed, chemistry on aerosol surface could be changed, besides the BC-radiation and BC-cloud interactions. To separate the role of BC and non-BC, I would suggest the authors do another parallel simulation with all aerosol emissions turned off. Or at least discuss the biases of the results.*

**Response:**

   We have conducted four parallel numerical experiments in this study. At first, the control experiment (EXP_CTL) is conducted with complete emission and with ARI+ACI turned on. Then, anthropogenic BC emission was removed from the model with both ARI+ACI still turned on (EXP_NOBC). After that, we conducted the experiments that turned off both ARI and ACI (EXP_NOAER), which has neither ARI nor ACI from the BC and non-BC aerosols at all. In EXP_NOAER, we shut off the aerosol impact on the radiation transfer and cloud as well as the wet removal and cloud chemistry, which make this experiment equivalent to a parallel experiment that remove all aerosols regarding the interacting between aerosols and fog formation. So, the effect of BC can be calculated as the EXP_CTL - EXP_NOBC, the effect of non-BC is EXP_NOBC - EXP_NOAER. This method well separated the role of ARI and ACI, and distinguished the role of BC from the non-BC with less computational cost. And the role of BC was later proved be an important finding in this article, which was to enhance the moisture advection mainly through its radiative effect during the fog-haze event in the early December 2013 in the YRD region. To clarify this, we will add more description about the experimental design in section 2.2. In EXP_CTL, the BC and other aerosol are assumed to be internally mixed within each bin. Without BC emission, in EXP_NOBC, the particle aerosol

would absorb much less shortwave radiation, since BC is the largest absorption particle in WRF (https://ruc.noaa.gov/wrf/wrf-chem/tutorial2018.htm, Page 11). This change due to BC's removal is mostly accountable for the cooling of the ambient atmosphere where high loadings of BC locate, thus indicating the heating effect of BC's role in this study. Not only that, the change to the mixing state of BC and other aerosols can also leads to the significant change to the optical property of the aerosol particle, which is not discussed in this paper. While, other changes, such as the change to the surface chemistry and the vegetation can also be important influencing factors. They are also not discussed in this work, since they do not affect the main mechanism for the advection-radiation fog that we investigated. However, this argument provides us some insight to the BC roles on affecting the fog in a more generally way, which needs more attention in the future work.

*Minor comments:*
*Page 1 Line 24: 'heave' -> 'heavy'*

**Response:**
   Accepted. We will correct it in the revision.

*Page 2 Line 15: In addition to impact on PBL, recent study also found BC can change land-sea thermal contrast and weaken the East Asian winter monsoon (Lou et al., 2019), which mechanism is very similar to this study, as well as large-scale circulations (Yang et al., 2019).*

*Lou, S., Y. Yang, H. Wang, S. J. Smith, Y. Qian, and P. J. Rasch, Black carbon amplifies haze over the North China Plain by weakening the East Asian winter monsoon, Geophys. Res. Lett., 46, 452–460, doi:10.1029/2018GL080941, 2019.*

*Yang, Y., S. J. Smith, H. Wang, C. M. Mills, and P. J. Rasch, Variability and timescales in the climate response to black carbon emissions, Atmos. Chem. Phys., 19, 2405-2420, doi:10.5194/acp-19-2405-2019, 2019.*

**Response:**
   Indeed, these references are quite related to this work. We will discusses and add these two new studies in the introduction and discussion part.

*Page 4 Line 26: Please clarify that how did the authors turn off ARI and ACI in the model. What variables they excluded or fixed in model?*

**Response:**
   According to user guide of WRF-Chem (https://ruc.noaa.gov/wrf/wrf-chem/Users_guide.pdf) page 26 - 27, both of ARI and ACI can be turned on and off by configuring the

namelist.input. The ARI is turned off by setting aer_ra_feedback = 0. The ACI is further turned off by setting the chem_opt to 201 (mosaic without aq), wet_scav_onoff = 0 (wet scavenging off), cldchem_onoff = 0 (cloud chemistry), and progn = 0 (turns off prognostic cloud droplet number), then the aerosol would have no impact on the cloud/fog droplet. The effect of the ACI calculated this way represented the ACI effect from the fog-haze event, which is reasonable to be insignificant compared with the ARI under heavy polluted condition.

*Page 5 Line 4: Please provide the sites or data reservoir where the authors got all these data.*

**Response:**

Here list the dataset used in this work. To make it more clear, we will add them in the acknowledgments.

1. NCDC dataset was obtain from https://gis.ncdc.noaa.gov/maps/ncei/cdo/hourly
2. The vertical profile sounding was retrieved from http://weather.uwyo.edu/upperair/sounding.html
3. The CNEMC dataset was gather from http://106.37.208.233:20035/ since 2013.

As for the sensible heat, shortwave radiation flux and BC concentration observation, they were retrieved from SORPES at NJU, Xianlin Campus, which we'd like to provide to those who are interesting on the subject.

*Figure 4: It should be days 2,5,6,"9".*

**Response:**

Accepted. Figure 4 will be corrected in the revised manuscript.

*Page 6 Line 22: underestimated during the occurrence of fog in Nanjing "in the model".*

**Response:**

Accepted. We will revise this sentence.

*Page 6 Line 26: a possible enhancement of the formation of secondary aerosols through aqueous phase or heterogeneous reactions "in the real world".*

**Response:**

Accepted.

*Page 7 Line 23-28: I am confused that why temperature over the sea was less sensitive to the reduction of incoming solar radiation but sensitive to aerosol warming effect? Both of them are*

*changes in radiative fluxes. Large heat storage capacity only implies that ocean is insensitive to heat or cooling effects (changes in radiative fluxes) compared to land. Instead, Lou et al. (2019) found that BC-induced heating over Bohai Sea, Yellow Sea, and East China Sea evaporated low cloud and increased high cloud, leading to larger warming over oceans east of China mainland.*

**Response:**

Since the clean atmosphere only absorbs a small fragment of the incoming solar radiation. The near surface layer, being close to the land/sea surface, is mostly warmed up during daytime by taking the sensible or latent heat that emitted from the surface. This process essentially transfers the solar energy into the internal energy of the underlying surface, and then pass it to the air near the surface. However, the water body has a large heat storage capacity than the land. That means the fact that the heating of the near-surface atmosphere would be slower over the water body (which is actually the main cause for the sea breeze). Thus, when the aerosols reduce the incoming solar radiation, the aerosol cooling effect on the near surface atmosphere is correspondingly less sensitive over the sea than that over the land. With the same amount of aerosol concentration increased over both land and sea, the near surface air temperature would have less reduction over the sea than the land surface. Part of the intercepted solar energy is absorbed by the particle like BC, if present, and warm the atmosphere directly. This heating effect from absorbing aerosol can be more efficient than taking the heat from the underlying water body during the daytime. Therefore, over the sea, the near surface temperature is generally more sensitive to the heating effect than the cooling effect of the aerosol. In our case, the BC's distribution over the sea surface and corresponding heating region matched well (Figure 10 and Figure S4), suggesting the heating effect from BC increase the air temperature in the near surface over the offshore region in the East China sea. Similar to our study, Wang et al. (2015) found as large as +0.3 °C feedback from aerosol heating effect over Gulf of Mexico and Atlantic Ocean and – 0.3 °C feedback over eastern U.S. during July 2006. As for the change to the cloud, the simulation result from both EXP_CTL (Figure R1a and b) show that the cloud cover over the eastern China is low before the formation of fog on 7 December. After, the BC is excluded from the model, less cloud cover condition merely changes (Figure R1c and d). And below or above the 850 hPa, the cloud cover or its change due to BC is much smaller. On the other hand, if BC change the cloud cover over the sea which lead to the heating over the surface, there would be a corresponding positive change in the Downward Shortwave Radiation (DSW). However, there're only negative change in the DSW (Figure R2e) in the heating region over the eastern China Sea (Figure R2b). By comparisons, impact from Total or non-BC indicate the increase of temperature over the East China Sea was mostly result from BC effect (Figure R2d and f). Therefore, in this study, we conclude the warming over the East China Sea is the direct result from the BC's heating effect over the sea for the early December 2013. This leads to the enhancement of moisture advection and consequently the increase of the moisture in YRD and adjacent offshore region in the near surface (Figure R2i) as discussed in the article.

[Figure]

**Figure R1.** The mean cloud water mixing ratio and cloud fraction at 850 hPa over the eastern China during LST 8:00-17:00 6 December 2013. (a) and (b) are results from EXP_CTL, while (c) and (d) are result from EXP_NOBC.

[Figure]

Figure R2. Average changes of DSR (upper), Temperature in the near surface (middle) and water vapor mixing ratio in the near surface (lower) induced by total effect of aerosols (left), effect of BC (middle) and effect of non-BC aerosols (right) in the daytime (08:00 – 17:00 LST) during 1-10 December 2013.

*Page 8 Line 30: The BC impact on surface temperature depends on vertical location of BC and also depends on models.*

**Response:**

Accepted. We will add more explanations to the response of near-surface atmospheric temperature to the radiative effect of BC being different to that of non-BC aerosols. And the statement of the uncertainties in simulating the BC impact on the near surface temperature in numerical modellings will be also included.

*Page 9 Line 2: What does "other forcings" mean.*

**Response:**

    Here, the "other forcings" is referred to the other cause of the RH increase rather than the decrease air temperature induced by local aerosol cooling effect. Here, it was indicated that the moisture advection from outside the YRD region might be the cause of the large increase in RH over the YRD region on 6 December 2013. We will change "other forcing" to "other factors such as moisture advection" in order to avoid ambiguous indications.

**References:**

Wang, K., Zhang, Y., Yahya, K., Wu, S. Y., and Grell, G.: Implementation and initial application of new chemistry-aerosol options in WRF/Chem for simulating secondary organic aerosols and aerosol indirect effects for regional air quality, Atmos. Environ., 115, 716-732, doi: 10.1016/j.atmosenv.2014.12.007, 2015.